# Equine Herpesvirus Type 1 Myeloencephalitis in the Brazilian Amazon

**DOI:** 10.3390/ani13010059

**Published:** 2022-12-23

**Authors:** José Diomedes Barbosa, André de Medeiros Costa Lins, Henrique dos Anjos Bomjardim, Natália da Silva e Silva Silveira, Camila Cordeiro Barbosa, Edsel Alves Beuttemmuller, Marilene Farias Brito, Felipe Masiero Salvarani

**Affiliations:** 1Instituto de Medicina Veterinária, Universidade Federal do Pará, Castanhal 68740-970, PA, Brazil; 2Instituto de Estudos do Trópico Úmido (IETU), Universidade Federal do Sul e Sudeste do Pará (Unifesspa), Xinguara 68557-335, PA, Brazil; 3Centro de Ciências Agrárias, Universidade Estadual de Londrina (UEL), Londrina 86057-070, PR, Brazil; 4Departamento de Epidemiologia e Saúde Pública (DESP), Instituto de Veterinária (IV), Universidade Federal Rural do Rio de Janeiro (UFRRJ), Seropédica 23890-000, RJ, Brazil

**Keywords:** infectious disease, horses, neurological clinical signs, nested PCR, serum neutralization

## Abstract

**Simple Summary:**

The horse breeding industry has great importance in the state of Pará, contributing to society in economic and cultural ways. For this, it is necessary to know the disorders that affect these animals, especially those caused by worldwide distributed virus, such as equine herpesvirus type 1. In this sense, the clinical and epidemiological investigation of this work about equine herpesvirus myeloencephalopathy show important results to the inclusion of this neurological disorder in the differential diagnosis of neurological diseases most common in horses in the state of Pará, Brazil. The work shows the viral circulation in different farms, clinical signs, epidemiological findings, macroscopic and microscopic findings in that animals submitted to necropsy.

**Abstract:**

An investigative and epidemiological study was carried out for equine herpesvirus type 1 (HVE-1) in 10 outbreaks of neurological disease from different farms in the state of Pará, Brazil. 25 horses were studied: six male and 19 females, aged between one and 13 years. A necropsy of six horses was performed, and the others recovered either with or without treatment (T1—vitamin B1 + dexamentasone; T2—vitamin B1 + flunixim meglumine). Animals that received treatment recovered after eight days. The main clinical signs observed were motor incoordination, progressive paresis, thoracic and/or pelvic limbs abducted after induction of clinical examination, knuckling of the hind fetlocks, sagging and swaying of the hindquarters while standing or walking and paresis. All animals were positive: 88% (22/25) in nested PCR and 72% (18/25) in serum neutralization (including three negatives in serology). Focal brownish areas compatible with hemorrhage were found in the white and gray matter of the spinal cord of two animals. On histological analysis, there were perivasculitis and neutrophilic vasculitis in the gray matter of the spinal cord and brain. Based on the evidence, this work proves the circulation of HVE-1 in the Amazon biome, mainly in the state of Pará, Brazil.

## 1. Introduction

Equine herpesvirus is a group of enveloped DNA viruses of the family *Herpesviridae*, subfamily *Alphaherpesvirinae* [1]. Five types of herpesviruses were identified as causing horses’ diseases [2]. Equine herpesvirus type 1 (EHV-1) and equine herpesvirus type 4 (EHV-4), genus *Varicellovirus* [3], are responsible for respiratory tract disorders in these animals [4]. However, EHV-1 is also implicated in reproductive disorders, like abortion [5] and in neurological disorders [6,7,8]. This group of viruses is enzootic in most of the world’s equine population [9] and is responsible for causing considerable economic losses in equine farming [10]. Economic losses occur mainly due to abortions and neonatal diseases [11], as well as expenses with the treatment of animals with respiratory and neurological clinical signs [12,13] or the death of animals [10]. 

Herpesvirus myeloencephalopathy is a disease that can develop in individual cases or outbreaks without predilection for age or sex [14] The establishment of infection usually occur within the first weeks of life [15] and outbreaks can appear when new animal are introduced into the herd [16] or by viral reactivation. Herpesviruses typically causes latent infections by long periods, then risk factors, such as housing and crowding, as well as stress, can trigger the disease, which is characterized by brain and spinal cord vascular lesions [17]. Latency is established in the trigeminal ganglion, olfactory nerve, sacral nerves and in the respiratory lymphoreticular system [10,17] Neurological disorders caused by EHV-1 and EHV-4 are poorly described in Brazil. The first occurrence of herpetic myeloencephalopathy caused by EHV-1 was described in the state of São Paulo in 2008 [18]. Since then, only a few isolated cases have been reported in the country [12,19,20,21,22]. In a retrospective study, 75 samples of central nervous system (CNS) obtained from horses with neurological disease were analyzed and it was found that 52% of them were positive for EHV-1 through the polymerase chain reaction assay (PCR) [20]. In this way, even as few reports in the literature, EHM should be a differential diagnosis of other diseases of horses with neurological clinical signs.

Because of the importance of herpetic myeloencephalopathy in equids as an emerging disease and the scarcity of reports of this disease in the national literature, it is necessary to include it in the differential diagnosis of neurological disorders in horses. Therefore, the objective of this work was describing the occurrence of EHM in the state of Pará, Amazon, Brazil.

## 2. Materials and Methods

### 2.1. Animals and Blood Sample Collection

The investigative and epidemiological study for EHV-1 was carried out in 10 neurological disease outbreaks with twenty-five horses (22 horses with neurological clinical signs, 3 horses apparently healthy) from 10 different farms located in the state of Pará, Amazon, Brazil. Blood samples were acquired by jugular venipuncture using vacuum tubes with and without EDTA (ethylenediamine tetraacetic acid) to obtain of whole blood and serum, respectively. Serum was obtained by centrifugation (2500× *g*, 10 min). Serum and whole blood were stored at −20 °C until laboratory procedures for molecular biology were performed. 

### 2.2. Serological and Molecular Diagnosis

The blood serum aliquots were submitted to serum neutralization test in cell cultures for the detection of antibodies of HVE-1 at the Biological Institute of the state of São Paulo, and the whole blood samples were submitted to the nested polymerase chain reaction (nested PCR) to detection of HVE-1. DNA collection was performed using the silica-guanidine isothiocyanate method, as described by Boom et al. [23]. The Nested PCR was performed following the recommendations of Borchers & Slater [24]. The first round of Nested PCR was performed with primers BS-1-P1 (5′-TCTACCCCTACGACTCCTTC-3′) and gB-1R-2 (5′-ACGCTGTCGATGTCGTAAAACCTGAGAG-3′). A second round of Nested PCR was used the primers BS-1P3 (5′-CTTTAGCGGTGATGTGGAAT-3) and gB-1R (5′-AAGTAGCGCTTCTGATTGAGG-3′), which amplifies a partial 771 bp. The examination was performed by 2% agarose gel electrophoresis stained with ethidium bromide and analyzed in a transilluminator UV-300 nm). To exclude other causes of neurological diseases, the differential diagnosis was performed: direct immunofluorescence, mice inoculation and histophatological analyses for rabies; PCR for EHV-4, pseudorabies virus (SuHV-1, porcine herpesvirus type 1), eastern equine encephalitis (EEEV), western equine encephalitis virus (WEEV), Venezuelan encephalitis virus (VEEV) and Saint Louis encephalitis virus (SLEV); histophatological for equine leukoencephalomalacia (LEME).

### 2.3. Animals Clinical Treatment

Two protocols were established for the clinical treatment of the animals during the outbreaks. In treatment 1 (T1), the animals received 10 mg/kg of vitamin B1 (intramuscular) and decreasing doses of 0.1 mg/kg of dexamethasone (intravenous) for ten days. In treatment 2 (T2), the animals received 10 mg/kg of vitamin B1 (intramuscular) for ten days and three applications of 2 mg/kg of flunixim meglumine (intravenous).

### 2.4. Necropsy and Histopathological Examination

A necropsy of six horses was performed, with a collection of fragments of several organs and of the central nervous system, that were fixed in 10% buffered formaldehyde for histopathological studies in the Pathological Anatomy Sector of the Federal Rural University of Rio de Janeiro (UFFRJ). Tissue samples were routinely processed, soaked in paraffin, cut into a microtome at 5 µm thickness and stained by hematoxylin and eosin (HE).

## 3. Results

Ten neurological disease outbreaks in horses, consistent with infection of HVE-1, were inspected on 10 farms located in five cities in the state of Pará, Amazon, Brazil. In these outbreaks, 25 horses (six males, 19 females) were examined, six males and 19 females; the age varied from one to 13 years. Twenty-two horses had clinical neurological signs and three animals were apparently healthy; of these, six horses died, and the others recovered with or without treatment (Table 1).

The main clinical signs observed are shown in Figure 1. Most of them were characterized by motor incoordination, progressive paresis, thoracic and/or pelvic limbs abducted after induction of clinical examination, knuckling of the hind fetlocks, sagging and swaying of the hindquarters while standing or walking and paresis, which at times was so severe that the animal almost dragged its hindquarters on the ground, on a steep slope, giving an appearance of a “crouching” animal (Figure 2a). Other clinical signs were observed less frequently, like tachycardia, tachypnea, tongue tone decreased, difficulty to get up, localized sweating, bedsores, recumbency with paddling limbs, muscle fasciculations in the face, frequent urination, priapism, limb spasticity, and dragging of the toe (Figure 2a). There were no reports of abortions or respiratory signs, and the animals were not vaccinated.

Eighteen animals showed a positive reaction for HVE-1 in serum neutralization, which corresponds to 72% (18/25). In Nested PCR there was detection of genetic material of HVE-1 in whole blood of 88% (22/25) of the horses tested. Three animals negative in the molecular method were positive in serology (Horses 2, 5 and 24). All animals studied in this work were negative for rabies, leukoencephalomalacia, EHV-4, SuHV-1, EEEV, WEEV, VEEV and SLEV. 

Horses that received treatment protocols one and two recovered after eight days. However, 28 days after the end of treatment, horse 19 had recurrence of clinical signs and died in 46 days, from the first clinical manifestation. Among the 25 animals studied, six horses died (Horse 1, 6, 13, 19, 20 and 25) and the macroscopic lesions of the nervous system consisted of focal brownish areas (Figure 3), suggestive of hemorrhage, in the spinal cord, in the white matter and gray matter of the spinal cord of two animals (Horses 1 and 25). In the other organs, there were no lesions, or the lesions found were few and unspecific (no direct correlation with the disease). In the histological analysis, horses 1, 19 and 25 presented characteristic lesions of meningitis and encephalitis: perivasculitis and neutrophilic vasculitis in the gray matter of the spinal cord and the brain (Figure 4), as well as hemorrhagic focuses in the white and gray matter of the spinal cord, which are detailed in Table 2.

## 4. Discussion

In the present study, a greater number of females were affected than males (19/25), in line with what has been observed in other studies [12,14]. However, Mekonnen et al. [25] described that there are no significant differences between sex. Mares are more likely to develop herpetic myeloencephalopathy [26], and the association with abortions is a likely explanation because this is a clinical sign usually associated with outbreaks [12]. More studies are needed to investigate the occurrence of this disease in females [14].

Horses over five years of age have a higher risk of progression to a neurological type when infected with EHV-1 [27] and a greater magnitude of viremia in elderly horses is an important risk factor of the development of neurological signs [28]. In the studies by Mekonnen et al. [25] and Negussie et al. [14], greater proportion of seropositive horses for EHV-1 were over three years of age and probably can be explained by one longer exposure title of the virus, an epidemiological factor also observed in this present work, in which horses with more than three years were most infected (19/25). Age was probably one of the main risk factors for the animals develop neurological signs in this study.

Although almost all animals showed some neurological clinical manifestation (88%–22/25), three of them did not show any clinical signs and were apparently healthy. According to the literature, this can be explained by the ability of EHV-1 to establish long latency periods in animals [17,29,30] and approximately 50 to 60% of recovered animals remain with the latent infection in the body [17]. Latent herpesvirus infections can be reactivated by stress, corticosteroids or other drugs [29], especially immunosuppressants. The presence of specific antibodies against EHV-1 is an important epidemiological finding, especially in clinically healthy animals, because horses with latent infections are asymptomatic carriers that can spread the virus during reactivation [30] and the introduction of asymptomatic animals into properties with susceptible animals can promote the emergence of outbreaks [16].

Cases of herpetic myeloencephalopathy in Brazil occur in isolation and throughout the year [20]. However, the outbreaks described in this work showed seasonal characteristics, with greater occurrence from December to March. Probably the high rainfall, high temperatures, high humidity and the increase of hematophagous insects may be one of the causes of stress, recrudescence and development of nervous symptoms in horses infected by EHV-1. Furthermore, the confinement of equids in certain periods is a risk factor, which may contribute to the seasonality of the disease [14]. 

EHV-1 infection usually presents respiratory signs, with abortions and neurological alterations, but some clinical cases present only nervous symptoms [16]. The clinical nervous signs presented by the animals in this report are related to the macroscopic and microscopic lesions evidenced in the brain and spinal cord. According to Ata et al. [30] the normal CNS functioning depends on the motor activity initiated through by stimuli originated in the higher motor centers located in the brain. These electric impulses are transmitted to muscle, then spinal cord and spinal nerves integrity is necessary. 

The main neurological clinical signs in this study were related to the pelvic limbs, which was also described in Brazil by Costa et al. [12], characterized by flaccid paralysis of the hind limbs, incoordination and ataxia. Different degrees of proprioceptive deficit, ataxia and pelvic limb paresis that progressed to lateral recumbency (severe cases) were also observed by Negussie et al. [14] and McFadden et al. [31]. Furthermore, according to the literature, horses infected by EHV-1 usually have severe and progressive ataxia [32,33]. On the other hand, less frequent clinical signs such as fever, lethargy, urinary incontinence and decreased consciousness were also described by Galen et al. [34] and Pusterla et al. [33]; of these, urinary incontinence and fever are frequently reported in horses infected by EHV-1 [8,10,31,35], which were not observed in this work.

Virus culture and isolation are the gold standard test for EHV-1 [15], but the isolation is difficult and highly variable [20]. On the other hand, PCR has been used as the test of choice, because of its high sensitivity and specificity [15]. It is described that in the acute phase of equine herpesvirus myeloencephalophaty, the diagnosis can be made by viral isolation or by PCR from nasopharyngeal swab, peripheral blood or cerebrospinal fluid [8,14,36]. Virus neutralizing antibodies can be used in outbreaks to find horses that were exposed to the virus and can provide presumptive evidence of EHV-1 infection. However, virus neutralization antibodies must be distinguished antibodies to EHV-1 and EHV-4 [15]. In the present study, 88% of the horses amplified DNA of EHV-1 (22/25), and three of them were asymptomatic. Similar results were obtained by Hafshejani et al. [37], who applied PCR for EHV-1 in blood samples from a population of ninety apparently healthy horses and obtained approximately 27% of positivity; the authors argued that positivity is indicative of viremia and, consequently, an active infection, which was also reported by Pusterla et al. [8], and the absence of clinical signs can be explained by the ability of animals’ immune system to suppress the virus. In this way, asymptomatic positive equines in the present study (horses 15, 16 and 24) can be considered carriers of EHV-1 and represent a risk of infection for other animals in the herd. 

Eighteen animals showed nervous symptoms and were treated with anti-inflammatory drugs and vitamin B1 (T1 and T2 treatments); 72% (13/18) of them showed clinical recovery, which demonstrates that these treatment protocols are effective for myeloencephalopathy by EHV-1 in horses in the Amazon biome. Anti-inflammatory drugs are able to decrease infection, probably by reducing the contact between infected peripheral blood mononuclear cells and endothelial cells in vitro [38]. In addition, mice experimentally infected with EHV-1 demonstrated a strong pro-inflammatory response and developed severe encephalitis [39]. Vitamin B1 acts as a coenzyme in carbohydrate metabolism, in protection against oxidative stress and in the nerve regeneration process [40]. Even after treatment, some animals may show viral reactivation and resumption of clinical signs within a few days [10], which was observed in 38% (7/18) of horses treated in this study, three of them died.

Evidence suggests that the administration of antivirals, such as valaciclovir, can decrease EVH-1 replication and clinical signs, especially if the treatment begins in the early stage of infection [32], before the onset of symptoms [8]. In addition, it has been reported that there may be an association of decreased risk of equine herpesvirus myeloencephalopathy in horses supplemented with zinc, which requires further studies in this regard [26].

Differential diagnoses were performed as described by Costa et al. [12], ruling out the occurrence of rabies virus, EHV-4, SuHV-1, EEEV, WEEV, SLEV and LEME. Additionally, rabies was also ruled out due to the following factors: evolution of clinical signs, which generally lasts an average of 10 within 12 days; non-involvement of other species and absence of Negri bodies in histological sections from the spinal cord. The epidemiology of the outbreaks ruled out the possibility of infection by *Sarcocystis neurona*, since there was no history of marsupials’ presence in the properties; in addition, the protozoan was not visualized in the histological sections of the central nervous system. There were no lesions characteristic of leukoencephalomalacia in the horses that died. Along with that, the pasture where horses grazed was extensively inspected and no specimens of *Bambusa vulgaris* (bamboo) were found. Furthermore, the satisfactory clinical response of the animals to the treatment by anti-inflammatory drugs and vitamin B1 was another important factor that helped to guide the diagnosis for HVE-1 infection.

Six animals were necropsied and three of them did not show important macroscopic lesions (horses 6, 13 and 20), which was also verified by McFadden et al. [31], who found no macroscopic changes in the nervous tissue of horses with equine herpesvirus myeloencephalopathy. However, there were areas of hemorrhage in the gray matter in the spinal cord, which extended slightly to the white matter and was different from the study carried out by Studdert et al. [41], who found hemorrhagic focuses from the white and gray matter of the brain, yellowish or brownish focal areas. In another work in Brazil, horses with equine herpesvirus myeloencephalophaty presented macroscopic alterations characterized by small hemorrhagic focuses scattered throughout the leptomeninges [20]. The histological alterations usually found in horses affected by the neurological form by EHV-1 infection are vasculitis and thrombosis of small-caliber vessels in the brain or spinal cord, as well as lymphocytic infiltrates and myelitis with axonal degeneration. But some animals may not show macro and microscopic changes [42]. The same pathological findings have been described in horses experimentally infected by EHV-1 and developed myeloencephalopathy [43].

## 5. Conclusions

The diagnosis of equine herpesvirus myeloencephalopathy in the state of Pará, presented in this study, was based on clinical-epidemiological findings, on the recovery of cases with or without treatment and the identification of the virus through the detection of antibodies or the virus by nested PCR. Pelvic limb paresis was the main clinical finding, while the main epidemiological finding was the occurrence of cases only in horses and in outbreaks. Microscopical lesions of the central nervous system were observed, showing that there was an active EHV-1 infection in the dead animals. Thus, the importance of including equine herpesvirus myeloencephalopathy as a differential diagnosis of neurological diseases that affect horses, especially in the Amazon Biome, is highlighted.

## Figures and Tables

**Figure 1 animals-13-00059-f001:**
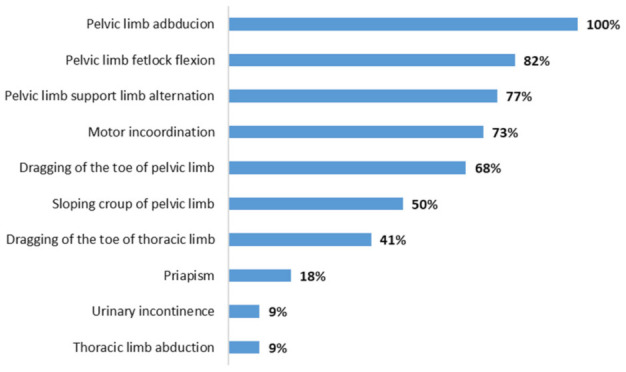
Main neurological signs in horses with equine herpesvirus myeloencephalopathy in the state of Pará, Amazon, Brazil.

**Figure 2 animals-13-00059-f002:**
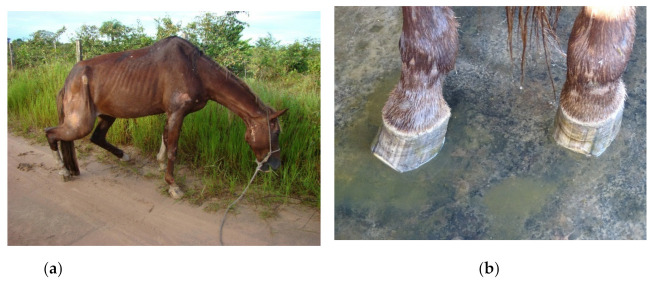
Clinical signs in horses with equine herpesvirus myeloencephalopathy in the state of Pará, Amazon, Brazil: (**a**) animal with sloping croup, abduction and pelvic limb paresis (“crouched posture”); (**b**) animal with moderate dragging of the toe of pelvic limbs, due to dragging on a concrete floor.

**Figure 3 animals-13-00059-f003:**
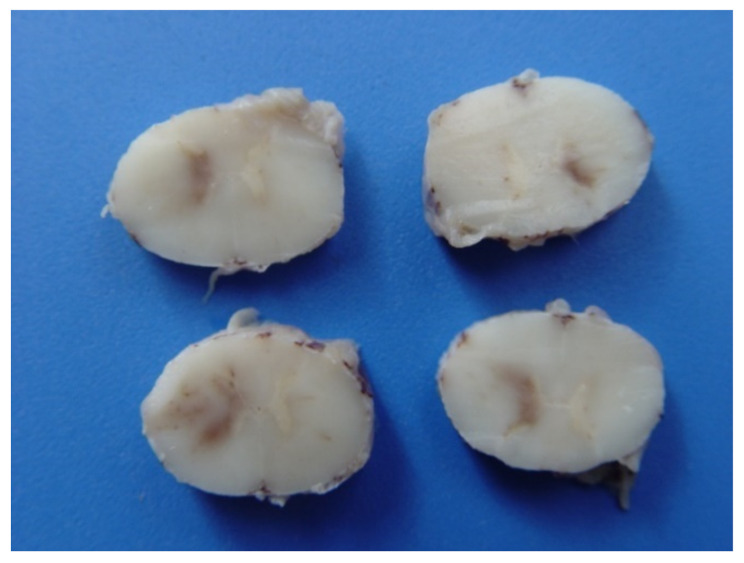
Hemorrhagic focuses in the gray and white matter of the spinal cord of an animal positive to equine herpesvirus myeloencephalopathy submitted to necropsy.

**Figure 4 animals-13-00059-f004:**
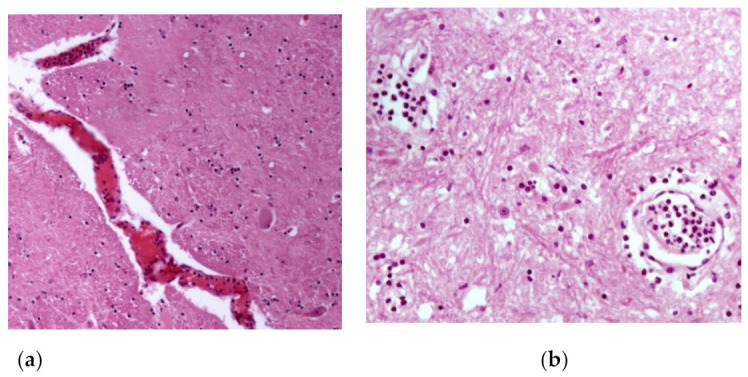
Microscopic changes observed of an animal submitted to necropsy (Obj. 20, H.E.): (**a**) congestion, perivascular edema and predominantly neutrophilic vasculitis in the leptomeninges of the spinal cord; (**b**) perivasculitis and neutrophilic vasculitis in the spinal cord gray matter.

**Table 1 animals-13-00059-t001:** Identification of animals and outbreaks, sex, age, presence of neurological signs, serology, nested PCR results and cases outcome of 25 horses suspected of equine herpesvirus myeloencephalopathy in the state of Pará, Amazon, Brazil.

Horses	Outbreak	Sex	Age *	Signs **	Serology	PCR	Treatment	Outcome
1	1	F	2	Present	P	P	NT	Death
2		M	9	Present	P	N	T1	Recovery
3		F	7	Present	N	P	T1	Recovery
4	2	F	4	Present	P	P	T1	Recovery
5		F	3	Present	P	N	NT	Recovery
6		F	5	Present	P	P	T1	Death
7	3	M	6	Present	N	P	TI^R^	Recovery
8		F	8	Present	P	P	TI^R^	Recovery
9		F	2	Present	N	P	NT	Recovery
10	4	M	9	Present	P	P	TI^R^	Recovery
11		F	9	Present	P	P	TI^R^	Recovery
12	5	M	1	Present	P	P	NT	Recovery
13		F	3	Present	P	P	T1	Death
14	6	F	14	Present	P	P	T1	Recovery
15		F	12	Absence	N	P	NT	NA
16		F	8	Absence	N	P	NT	NA
17	7	F	12	Present	P	P	T2	Recovery
18		F	12	Present	N	P	T2	Recovery
19	8	F	2.5	Present	P	P	T2 ^R^	Death
20		F	13	Present	P	P	T2 ^R^	Death
21	9	M	7	Present	P	P	T1	Recovery
22		M	4	Present	P	P	T1	Recovery
23	10	F	8	Present	N	P	T2	Recovery
24		F	12	Absence	P	N	NT	NA
25		F	7	Present	P	P	T1 ^R^	Death

M = male; F = female; P = positive; N = negative; T1 = treatment 1 (Vitamin B1 + Dexametasone); T2 = treatment 2 (Vitamin B1 + Flunixim meglumine); ^R^ = recrudescence after treatment; NT = no treatment; NA = not applicable; * Years of age; ** Neurological signs.

**Table 2 animals-13-00059-t002:** Necroscopic and histopathological findings of horses with equine herpesvirus myeloencephalopathy in the state of Pará, Amazon, Brazil.

Horse	Macroscopic Changes	Microscopic Changes
1	Brownish focal areas (suggestive of hemorrhage) in the gray matter of the spinal cord.	Spinal cord: congestion +(+) and neutrophilic leukocytosis inside and around meningeal and gray matter vessels, sometimes accompanied by fibrin inside vessels; presence of rare axonal spheroids; focal gliosis +(+).
19	Petechiae and ecchymoses +(+) in bladder; bladder ulcer with 4 cm of diameter with raised and irregular edges and surface covered by fibrin.	Spinal cord: activation of endothelial cells in meningeal vessels; rare lymphocytes and eosinophils inside meningeal vessels, sometimes leukocytostatis of neutrophils and fibrin; gliosis focuses (+); few axonal spheroids.
		Brain cortex: leukocytostasis of neutrophils in meningeal vessels; endothelial cell activation in meningeal vessels;
		Lung: diffuse edema +++; diffuse congestion ++; presence of shock bodies.
25	Brownish focal areas (suggestive of hemorrhage) in the gray matter ++ and white matter + of the spinal cord with meningeal hyperemia.	Spinal cord: endothelial cell activation in meningeal vessels; leukocytostatis neutrophils focuses on the meninges and white matter; rare axonal spheroids in white and gray matter.
		Hippocampus: gliosis focuses +.
		Thalamus: endothelial cell activation and leukocytostatis of neutrophils; areas of perivascular edema; neutrophilic vasculitis.
		Brain cortex: vasculitis focuses composed by neutrophils and macrophages; endothelial cell activation in gray matter vessels.
		Cerebellum: areas of perivascular edema.
		Lung: diffuse congestion ++; neutrophil leukocytoesterase in interstitial and alveolar vessels.
		Liver: circulating neutrophils ++, especially in centriolobular veins; pigment.

(+) = mild lesion; + = mild injury; +(+) = mild to moderate injury; ++ = moderate-intensity injury; +++ = severe-intensity injury.

## Data Availability

Not applicable.

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
