# Peer review of "Equine Herpesvirus Type 1 Myeloencephalitis in the Brazilian Amazon"

_animals, 2022, doi:10.3390/ani13010059_

Round 1

Reviewer 1 Report

Line 21 - ... findings in that animals

Line 57 - ...”EHV-1 and EHV- are poorly”...  it is missing  the genotipe of the second HVE

Line 82 - the correct writting is :...” submitted to serum neutralization”...

Line 83 – the correct word is - “antibodies”

Line 91 –the correct word is “agarose”

Describe better the methods used to the differential diagnosis on paragraphs 93 to 97

Line 114- change 6 by “six”

Line 152 - horse 19 who had recurrence of clinical sign and died:  this animal was positive to Equine Infectious Anemy Virus?

Page 6 (table 2)– horses 1, 19 and 25: could you review the therm ”neutrophils leukocytostatis” in the first line of the table 1? It seems to be more apropriate to use the therm “neutrophilic leukocytosis”

There was abortion in the mares?

Lines 187 and 188-  it should be better to write : ...“ can be explained by one  longer exposure title of the vírus”...

Line 2322- correct ...”Pusterla et al. [8],”...

Item CONCLUSIONS – it  is importante you say that microscopical lesions  of the CNS were observed, showing that there was an active HVE-1 infection in the dead animals.

Line 322 – write in italic “Equus caballus”

Lines 325, 373, 350 – “Fava, C. del “shold be substituted by “Del Fava, C.”

Line 374- the correct name of the jornal is “Rev. Educ.

Author Response

Dear reviewer

Best regards!

Reviewer 2 Report

1. Line 25: replace ter with either
2. Line 30: Please discuss the use of discuss use of serum neutralizing antibody testing when faced
with an outbreak of this or any disease in this manuscript

3. Line 52: is somewhat confusing. Is the point that EHV-1 virus is associated with long periods of
latency?

4. Line 82 needs words added add (for, antibodies) ...submitted for serum neutralizing antibodies...
the statement ...in cell culture... in this line is confusing

5. Line 111: following the gender descriptions in results do the authors have any thoughts on why
there was a much higher incidence in female horses and can they comment in the discussion
about whether the age had any impact?

6. Line 129 ...giving and appearance should be giving an appearance

7. Line 149 were all horses negative for EHV or for EHV-4?

8. Lines 177 and 178 should include in line ... with... what has been observed no observer

9. Line 186 change greated to greater

10. Line 188 change ... as to an...

11. Line 198 change carries to carriers

12. Lines 208 and 209 are confusing in ...but may also presents should likely read may also present
as...

13. Line 228 ... was should be changed to were

14. Line 277 founded should be found

15. Line 285change bases to based

Recommend publication with minor changes and some expansion in the discussion regarding usefulness
of neutralizing antibody titers versus PCR and virus isolation

Author Response

Dear Reviewer

Best regards!
